# The Role of Lifestyle Intervention, in Addition to Drugs, for Diabetic Kidney Disease with Sarcopenic Obesity

**DOI:** 10.3390/life12030380

**Published:** 2022-03-06

**Authors:** Shu-Hua Chen, Yao-Jen Liang

**Affiliations:** 1Nephrology Department, MacKay Memorial Hospital, New Taipei City 25160, Taiwan; meinyu@gmail.com; 2Department of Life Science, Graduate Institute of Applied Science and Engineering, Fu-Jen Catholic University, New Taipei City 24205, Taiwan

**Keywords:** sarcopenic obesity, muscle mass, fat mass, diabetes, chronic kidney disease, dual energy X-ray absorptiometry

## Abstract

Diabetic kidney disease is the leading cause of end-stage renal disease in developing and developed countries. The growing prevalence and clinical challenges of sarcopenic obesity have been associated with the frailty and disability of diabetic kidney disease. It has been reported that insulin resistance, chronic inflammation, enhanced oxidative stress and lipotoxicity contribute to the pathophysiology of muscle loss and visceral fat accumulation. Sarcopenic obesity, which is diagnosed with dual-energy X-ray absorptiometry, is associated with worse outcomes in kidney disease. Growing evidence indicates that adherence to healthy lifestyles, including low protein diet, proper carbohydrate control, vitamin D supplement, and regular physical training, has been shown to improve clinical prognosis. Based on the higher risk of sarcopenic-obesity-related renal function decline, it has led to the exploration and investigation of the pathophysiology, clinical aspects, and novel approach of these controversial issues in daily practice.

## 1. Introduction

The prevalence of end-stage renal disease (ESRD) is up to 10 times higher in people with diabetes than in non-diabetic individuals. Glycemic control continues to be focused on the targeting of glycohemoglobin (HbA1C) among individuals with diabetes and chronic kidney disease (CKD) over the last decades [1]. Despite intensive medical therapies, there remains a significant residual risk of diabetic kidney disease onset and progression. Diabetic nephropathy begins as glomerular hyperfiltration with increased glomerular filtration rate (GFR), and then GFR begins to normalize for several years. As a result of progressive metabolic and hemodynamic changes of glomerulus, renal injury is characterized by microalbuminuria (between 30–300 mg/day) over time. It is of note that the maximal benefits of good glycemic control occur prior to the onset of macroalbuminuria (≥300 mg/day), which are inevitable to progress to renal failure [2]. Nephrotic syndrome (≥3 g/day), elevated serum creatinine level, low serum albumin, hyperlipidemia, edema and hypertension precede ESRD, on average, by about 3 to 5 years, but this timing is extremely variable [3]. According to the USRDS ESRD available database, the percentage change in the incidence of treated ESRD attributed to diabetes appears to be strongest in Asia [4].

The pathogenesis of diabetic kidney disease (DKD) begins with vascular endothelial damage, mesangial cell proliferation and matrix expansion due to hyperglycemic hyperfiltration, advanced glycosylation of tissue protein and cytokines release, e.g., interleukin (IL)-1β, IL-6, tumor necrosis factor-α (TNF-α), and transforming growth factor-β (TGF-β) [5]. Pathologic nodular glomerulosclerosis appears to correlate with following tubular atrophy and interstitial fibrosis. Furthermore, other hemodynamic factors, such as vasoconstrictors, the renin-angiotensin system, endothelin-1, and vasodilators, prostacyclin (PGI_2_) and nitric oxide, are responsible for oxidative stress and the hyalinosis of afferent and efferent arterioles as well as arteriosclerosis. Advanced kidney disease is characterized by metabolic acidosis, electrolyte imbalance and uremia clinically, which promote excessive protein degradation, anorexia, muscle loss, edema and body weight loss. To increase awareness of the comorbid catabolic/anabolic alterations of chronic disease, it is important to identify the major pathophysiologic issues that show the way to minimize disability and slow down complications and comorbidities.

Metabolic disorders are common in diabetic kidney disease, such as chronic inflammation, oxidative stress, malnutrition, physical inactivity, muscle depletion and high body fat. In view of the upcoming aging and the increasing sedentary behavior, progressive accumulation of adipose tissue and impairment of muscle quantity and quality have been implicated as both a cause and consequence of altered glucose disposal, as skeletal muscle accounts for more than 80–90% of glucose clearance during hyperinsulinemia-euglycemic clamps [6]. It has become apparent that human skeletal muscle is an endocrine organ, which can secrete many myokines for the regulation of either autocrine, paracrine or endocrine actions. At a relatively lower mean body mass index (BMI) compared with those of European descent, East Asians have a greater amount of total body fat mass and a growing tendency to visceral adiposity, which increases metabolic risk [7].

In this review article, we focus on early the recognition of negative changes of body composition in diabetic nephropathy, and setting up its clinical algorithm of diagnosis and management.

## 2. Early Detection the Sarcopenic Obesity in Diabetes Kidney Disease

Diabetic nephropathy is asymptomatic in its early stages following the development of microalbuminuria. Progressive tubulointerstitial fibrosis has been proved to correlate with the magnitude of proteinuria. If left untreated, urine filtered albumin itself may damage proximal tubular cells by the synthesis of endoplasmic reticulum (ER) stress-related protein, e.g., caspase-12, and the accumulation of misfolded protein, e.g., glucose-regulated protein 78 (GRP78) and oxygen-regulated protein 150 (ORP150) [8]. Caspase-12, in turn, is responsible for the apoptosis of the cultured kidney epithelial cells (NRK-52E) by its potential regulation of nucleotide-binding domain-like receptor protein 3 (NLRP3) inflammasome with bovine serum albumin [9]. In addition, the upregulation of NLRP3 inflammasome leads to the processing of caspase-1 and the secretion of the proinflammatory cytokines interleukin (IL)-1β and IL-18 with neutrophil infiltration into kidney tissues of mouse models [10]. Uremic toxin indoxyl sulfate also contributes to ER stress and reactive oxygen species (ROS) production in cultured human proximal tubular cells, demonstrated by the increase in C/EBP homologous protein (CHOP) in Western blot [11]. In addition to reduced GFR with disease progression, overt proteinuria both directly and indirectly increases other organs’ damage. A cross section study of the general population in Japan revealed a significantly positive correlation between the number of components of the metabolic syndrome and the corresponding prevalence of microalbuminuria (*p* < 0.001) [12]. In reality, the Heart Outcomes Prevention Evaluation (HOPE) study showed that increased quartile of albuminuria is extremely associated with cardiovascular disease and all-cause mortality independently of traditional cardiovascular risk factors in patients with type 2 diabetes [13]. Furthermore, the reduced clearance of insulin or glucose-lowering medicine may lead to untoward hypoglycemia and metformin-associated lactic acidosis in advanced CKD per se. Early detection and intervention of kidney disease in diabetes, complications and comorbidities can be effectively slowed down.

Increased muscle protein catabolism and the obesity-related downregulation of adiponectin are also common among frail persons with progressive diabetic kidney disease. The coexistence of these sarcopenia and obesity has an extremely high risk of metabolic abnormalities, hyperglycemia and chronic inflammation. Fukuda et al. reported that eGFR significantly went to a more than 30% decline in people with type 2 diabetes and sarcopenic obesity, which was evaluated by dual-energy X-ray absorptiometry (DXA), in a retrospective observational study [14]. In the National Health and Nutrition Examination Survey (NHANES), there was underestimation of obesity by BMI (41% obese) compared with by DXA (71% obese) in adult participants with estimated glomerular filtration rate (eGFR) of 15–29 mL/min per 1.73 m^2^ [15].

There are various pathophysiological changes that link metabolic abnormalities and muscle deficits. Increased insulin resistance, free fatty acid, inflammatory cytokines, advanced glycosylated end-products (AGE), and decreased mitochondrial oxidative capacity-related lipid accumulation are associated with subsequent kidney disease. Following glucose intolerance and diabetes, insulin resistance plays a relevant role in impaired glucose transport by reduced insulin receptor tyrosine kinase activity, diminished glucose mitochondrial oxidative phosphorylation for generating adenosine triphosphate (ATP) and reduced glycogen synthase [16]. It is also well established that insulin is a potent inhibitor of lipolysis from adipocytes. The chronic elevation of plasma fatty acid concentration is a markedly causative role in the insulin resistance of skeletal muscle and results in lipotoxicity in kidney, heart and other organs, which lead to inflammation, cellular dysfunction and death [17]. Adiposity may be a potential risk factor for the development of kidney disease due to upregulated renal plasma flow, intraglomerular pressure, and renin-angiotensin-aldosterone system activity. The accumulation of AGEs has been identified in muscle atrophy and poor regenerative capacity in mouse and human myoblasts. AGEs reduced myotube diameters in vitro with a dose = dependent manner [18]. Hyperglycemia is also correlated with increased inflammatory markers, including tumor necrosis factor-α (TNF-α), interleukin (IL)-6 and C-reactive protein (CRP). TNF-α has been shown to impair insulin signal in peripheral tissues. It is well known that elevated serum TNF-α and leptin in type 2 DM are associated with obesity. On the contrary, IL-6 has been reported to cause muscle atrophy. In a longitudinal prospective aging study in Amsterdam in the elderly population, it was shown that high serum IL-6 (>5 pg/mL) and CRP (>6.1 mug/mL) levels were involved in two-to-three-fold greater risk of muscle strength loss [19]. There are novel explorations of inflammation factors that mark frailty and predict complications in diabetic subjects, such as neutrophil-to-lymphocyte ratio (NLR) and platelet-to-lymphocyte ratio (PLR). A retrospective study showed NLR and PLR are positively correlated with lower extremity vascular lesion in diabetes with available ankle-brachial index data and superior predictive ability achieved by PLR [20]. Furthermore, different levels of NLR and PLR have substantial influences on the survival of frail patients with maintenance hemodialysis [21].

## 3. Diagnosis of Sarcopenia Obesity

Sarcopenia has just been referred to as loss of skeletal muscle mass and strength that both accrue across a lifetime span and lead to difficulty with daily living activities. In CKD, sarcopenia is not just age-related change; it occurs as a result of the negative net protein balance from the disease as well, especially from the dialysis session in itself, either hemodialysis or peritoneal dialysis. Foley et al. have verified the correlation between increased sarcopenia prevalence and declining glomerular filtration rate based on bioimpedance measurements in community indwelling adults [22]. Traditionally, muscle wasting is common among persons with lower eGFR and high BMI may be protective in patients with CKD. In that regard, an accurate understanding of how to measure body composition beyond BMI affects outcome variables of a nutritional disturbance known as protein-energy wasting in CKD. DXA is a well-defined tool to accurately classify sarcopenia and obesity with a low X-ray beam for noninvasive assessment of muscle quantity and fat mass. The updated European Working Group on Sarcopenia in Older People (EWGSOP2, version 2019) adopted appendicular skeletal muscle mass (ASM)/height2 based on DXA, with cut-off points for males <7.0 kg/m^2^ and females <5.5 kg/m^2^ to define sarcopenia [23]. Applying the EWGSOP2 diagnostic criteria, reduced chair stand capacity (time to perform five repeated chair stands >15 s), gait speed test ≤0.8 m/s, or reduced grip strength (<16 kg for women and <27 kg for men) are advised as indicators of severe sarcopenia.

Moreover, the combined impact of sarcopenia and obesity, i.e., sarcopenic obesity, represents a double burden and is vulnerable to aging, physical inactivity, chronic inflammation, and metabolic syndrome. To define obesity, cut points of BMI ≥ 30 kg/m^2^ (corresponding values of ≥27 kg/m^2^ in Asian/Middle East/Mediterranean populations), fat mass > 42% in women and >30% in men associated with age-specific BMI on DXA, or central obesity as a waist circumference ≥88 cm in women and ≥102 cm in men (corresponding values of ≥80 cm in women and ≥94 cm in men in Asian/Middle East/Mediterranean populations) are used. Android-to-gynoid fat mass ratio (A/G ratio) can also be used as a surrogate marker for visceral fat accumulation: >0.80 for men and >0.62 for women. In fact, the combined impact of sarcopenia and obesity represents a double burden and is vulnerable to aging and chronic disease. A systematic review conducted showed that sarcopenic obesity is believed to account for 38% more risk of type 2 diabetes than those healthy control groups [24]. Additionally, this study revealed that nearly half of all adults with high prevalence of sarcopenia were overweight and had obesity without being influenced by their gender.

## 4. Relationship between Sarcopenic Obesity and Kidney Disease

Several non-inflammatory factors promote muscle wasting in kidney disease. Metabolic acidosis impairs the insulin receptor signal-transduction cascade that normally regulates glucose uptake in the skeletal muscle and suppresses proteolysis. Low pH also acts as a potent stimulator of protein degradation by triggering intracellular ubiquitin proteasome systems (UPS) and caspase-3 in CKD with acidosis [25]. The renin-angiotensin-aldosterone system (RAAS) plays a central role in hypertension and CKD. RAAS activates signaling pathways that maintain physiological balance of sodium and fluid, and regulates the homeostasis of blood pressure and cardiovascular circulation. Experimental evidence in animals suggests that angiotensin II reduced skeletal muscle regeneration via the inhibition of muscle stem (satellite) cell proliferation and the suppression of differentiation markers after cardiotoxin-induced muscle injury in vivo and in cultured satellite cells in vitro [26]. Recent growing novel findings in humans, in turn, recognize that changes in RAAS function are associated with metabolic changes in muscle quantity and fat mass. Clinical and epidemiologic research over the past two decades has witnessed a remarkable concept of the interaction of protein degradation and myostatin (MSTN)/activin system in muscle wasting of CKD. Myostatin, released by myocytes and as a negative regulator of muscle growth determining both muscle fiber number and size, has highly conserved autocrine function to inhibit myogenesis. In line with recent clinical studies, qualitative research has investigated the efficacy of MSTN/activin pathway antagonists in sarcopenic patients. Vitamin D insufficiency arises at an early stage of the renal disease as a result of decreased expression of 1-α-hydroxylase enzyme. Additionally, numerous epidemiological studies suggest that obesity is one of the risk factors for the exacerbation of vitamin D insufficiency due to its sequestration in the large pool of body fat. In a similar way, vitamin D deficiency is involved in the progressive loss of renal function. There is clear evidence that vitamin D deficiency, in turn, is related to insulin resistance in obesity, which impaired glucose transport and promotes hepatic glucose production, and has been associated with the risk of onset and progression of type 2 diabetes mellitus [27]. Some human and rodent studies support that vitamin D has been shown to have direct effects on reducing adipose tissue inflammation and improving hepatic insulin resistance. Low vitamin D level is an independent predictor for sarcopenia with low handgrip strength tests in elderly men. Early screening and replacement of vitamin D in frailty need to be further investigated [28]. On the other hand, there is a mouse model revealing that activation of vitamin D receptors by 1,25-dihydroxyvitamin D [1,25(OH)_2_D] upregulates the transcription expression of genes involved in calcium handling and muscle protein synthesis [29].

Notably, the accumulation of protein-bound uremic toxins, indoxyl sulfate (IS) and p-cresyl sulfate (PCs) has been associated with progression of kidney disease and systemic inflammation. IS derives from gut tryptophan metabolism and is eliminated by renal proximal tubules. Rodrigues et al. have shown that IS has direct toxic effects on myoblast by the induction of cell apoptosis of murine cell lines dependent on three different IS concentrations in vitro study. However, no effect was observed on the markers of myoblast differentiation, MyoD and myogenin mRNA expression by real-time PCR, at any IS concentration [30]. PC is formed by gut bacterial fermentation of tyrosine and phenylalanine and has previously been described to promote the redistribution of total body fat and modify insulin resistance. A cross section study showed a positive association of PCs and protein-energy wasting with higher inflammatory markers of LogIL10 and logIL12p70 in sarcopenic patients with advanced CKD [31].

Studies have shown that obesity is a potent risk factor for people developing de novo CKD. Obesity-related glomerulopathy (ORG) has been well defined with the property characteristics of glomerular hypertrophy, maladaptive podocyte depletion, focal segmental glomerulosclerosis, vacuolated tubules epithelium and interstitial fibrosis [32]. Subnephrotic proteinuria is the most common clinical presentation of ORG. Intriguingly, high metabolic rate tissues, such as heart myocardium, skeletal muscle and kidney tubule epithelium, preferentially take up and oxidize free fatty (FA) acid as an energy source during caloric expenditure. However, intracellular lipids overload and diminished β-oxidation of FA have been linked to net ROS production in mitochondrial and the subsequent apoptosis of proximal tubular epithelial cells. Thus, several lines of evidence support to enhance FA utilization can improve histopathology and slow CKD progression. Peroxisome proliferator-activated receptors (PPARs) proteins belong to the superfamily of nuclear transcription factors activated by agonists. There are three PPAR isoforms: PPARα, -β/δ, and -γ, encoded by separate genes. PPARα is abundantly expressed in the liver, intestinal mucosa, renal cortex (proximal tubules, medullary thick ascending limbs and glomerular mesangium), skeletal muscle and heart. PPARα is primarily involved in the β-oxidation of FA, lipolysis, generation of ketone bodies and regulation of energy storage in these areas of concern. Upon skeletal muscle as a major metabolic organ, it has also been known for its properties in most glucose and lipid homeostasis with PPARs agonists making some positive contributions. Fibrate, a potent PPARα agonist, was shown to trigger the expression of β-oxidation enzymes, long-chain and medium-chain acyl-CoA dehydrogenase, and acyl-CoA oxidase in rat renal cortex and to reduce renal lipid accumulation and the oxidative stress of glomerular and tubulointerstitial tissues [33]. There have been studies demonstrating that PPARα plays a key role in lipid metabolism, such as lowering triglyceride and raising high density lipoprotein levels. In contrast, PPARγ is highly enriched in adipose tissue, while lower distributed in the renal distal medullary collecting ducts, muscle and other tissues. The upregulation of PPARγ is associated with potential benefits on fatty acid metabolism and the expression of IL-6, TNFα and adiponectin. Interestingly, IL-6 is not only a cytokine, which was involved in pro-inflammation, but also a myo- and adipo-kine (“batokine”) with metabolic and regenerative activity. Studies in models of exercise/skeletal muscle contraction reveal somewhat myogenesis and insulin-sensitizing role of IL-6 following exercise [34]. In particular, IL-6 released from brown and beige adipose tissues contributes to whole body energy expenditure [35]. Thiazolidinediones-class drugs are the most extensively studied PPAR-γ ligands. PPARγ agonists have been shown to reduce blood glucose and insulin resistance, by comparing the effect of TNF-α in adipocytes, with potential therapeutic targets for obesity, dyslipidemia and diabetes. Recent studies have demonstrated beneficial effects of PPARγ agonists on reducing renal fibrosis and glomerulosclerosis by the inhibition of TGF-β-induced fibronectin expression in glomerular mesangial cells [36]. Moreover, PPARγ agonists significantly mediate protective effects against renal ischemia and reperfusion injury by the inhibition of expression intercellular adhesion molecule-1 (ICAM-1) and the reduction of polymorphonuclear (PMN) cell infiltration into rat renal tissues in vivo [37] (Figure 1).

## 5. Intervention of Sarcopenic Obesity in DM Nephropathy

With population growth and aging, the prevalence of diabetes and subsequent DM nephropathy continue to rise in the foreseeable future. Accelerated protein wasting and lipid accumulation occur from the advanced disease per se. There is ongoing development of novel therapeutic agents focused on skeletal muscle catabolism and anabolic dysfunction, such as testosterone replacement, selective androgen receptor modulators and anti-myostatin therapy. Age-related androgen depletion and lower testosterone level are known to be risk factors for various diseases, such as osteoporosis and sarcopenia. Further research is required on the development of testosterone therapy and androgen supplementation as promising agents in sarcopenia for less conclusive human data and concerns over cardiovascular and prostatic risks with a high dose. In a randomized placebo-controlled trial of testosterone supplementation for 6 months to older men with obesity, it was suggested that testosterone may attenuate the weight loss-induced reduction in muscle mass and hip bone mineral density [38]. Results similar to another randomized placebo-controlled trial of 5 mg/d of transdermal testosterone gel or placebo in older, frail men for 12 to 24 months, there was an increase in lean mass and a decrease in fat mass in the testosterone group, but no available effects in strength or physical performance [39]. In addition, various myo-statin inhibitors involving monoclonal antibodies, myostatin propeptides, soluble activin receptors and endogenous antagonists, and follistatin were provided in clinical trials for muscle hypertrophy and the reversal of cachexia [40].

Inadequate nutritional status due to protein-calorie malnutrition has been associated with the worsening of renal function. The main dietary interventions to prevent sarcopenic obesity in DM nephropathy include carbohydrate and protein restriction, proper amount of fat, and micronutrient supplementation. Low dietary protein, which results in a proportional reduction in urea generation and intraglomerular pressure, is routinely proposed. Moreover, the loss of amino acids, such as L-carnitine, and calories during hemodialysis and peritoneal dialysis promotes malnutrition and protein-energy wasting. L-carnitine deficiency, a nutrient in red meat, results in a decline in muscle power and fatigue, while L-carnitine supplement is effective for a decreased muscle mass in the elderly [41]. Limited saturated fatty acids (<7% of energy) and trans-fats have remained as major suggestions for atherogenic dyslipidemia. Their substitution with consuming plant oils and omega-rich fatty acid-containing oils in moderation is crucial to prevent malnutrition when total calories from other energy and macronutrients are restrained [42]. As most excitable cells, muscle fibers responding to ATP-driven sodium-potassium pumps maintain the resting potential or depolarization of muscle sarcolemma. As an interesting side note, a low-salt diet and restricted potassium intake are highly recommended in advanced CKD. A meta-analysis of randomized clinical trials on the benefits of dietary salt restriction has shown significantly reduced proteinuria in patients with CKD. The main outcome of urinary sodium excretion was 104 mEq/day and 179 mEq/day in low- and high-sodium intake subgroups, respectively. Mean differences in proteinuria were −0.39 g/day and −0.05 g/day [43].

Low levels of vitamin D have been proposed to be associated with insulin resistance. A cross-sectional cohort study demonstrated 25-hydroxyvitamin D (25(OH)D) serum concentration is inversely associated with several inflammatory biomarkers, such as high-sensitive C-reactive protein (hsCRP), TNF-α and IL-6, in severe obese individuals with mean BMI of 43.6 ± 4.3 kg/m^2^ [44]. However, the effectiveness of vitamin D supplementation on inflammation in patients with obesity was still controversial. Mousa et al. conducted a double-blind randomized trial and showed no effect of vitamin D supplementation, with a single 100,000 IU bolus followed by 4000 IU daily cholecalciferol or matching placebo for 16 weeks, on any inflammatory markers, such as hsCRP, TNF, monocyte chemoattractant protein-1 (MCP-1), interferon-γ (IFN-γ), several interleukins, and nuclear factor kappa-B (NFκB) in obese patients with BMI ≥ 25 kg/m^2^ [45]. Yet, the vitamin D receptor is expressed in muscle tissue and vitamin D supplementation was associated positively with physical performance and muscle strength in CKD patients with vitamin D deficiency (a serum 25-OHD level <20 ng/mL) [46]. Additionally, there were potential therapeutic issues for uremic toxin-targeted muscle atrophy of CKD. In conventional medical therapy of uremic toxins, it is mainly focused on controlling underlying disease, such as diabetes, hyperlipidemia and hypertension. To date, a growing number of studies are being designed to inhibit protein-bound toxins to alleviate CKD-associated skeletal muscle atrophy. A randomized controlled trial showed AST-120, an oral charcoal adsorbent of IS, had potential effects on gait speed change and quality of life, even though there was no significant difference in gait speed ≥0.1 m/s in 48 weeks [47]. It should be pointed out that exercise training has been effective in improving muscle strength in obese patients. Aerobic exercise improves endurance capacity and cardiopulmonary fitness, and anaerobic exercise regimens improve the cross-section area (CSA) of muscular fibers. An 8-week progressive resistance exercise training, consisting of 3 sets of 10 to 12 leg extensions thrice weekly, has been assigned to patients with CKD by Watson et al. Patients in the exercise group had a mean 8.3% increase in anatomical CSA of the midsagittal plane of the thigh using B-mode 2D ultrasonography with the individual prone at a 45° angle, with no change in the control group [48] (Figure 2).

## 6. Discussion

Sarcopenic obesity is associated with chronic low-grade inflammation, increased insulin resistance, and adipocyte hypertrophy. To date, some proinflammatory cytokines are capable of inducing muscle dysfunction by auto/paracrine manner, including IL-1, IL-6, IL-8, IFN-γ, TNF-α, MCP-1 and NFκB. In vitro studies showed insulin resistance and lipotoxic environment were characterized by intramuscular lipids accumulation with mitochondrial dysfunction, impaired β-oxidation capacity and in-creased reactive oxygen species formation. While reducing obesity is effective in reducing inflammation and delaying disease onset and progression, including diabetes, CKD and metabolic syndrome, the metabolic effects of hyperglycemia also lead to production of AGE, TGF-β and protein kinase C. These toxic products have some role in nephropathy and clinical albuminuria with histological changes of thickening of glomerular basement membrane, mesangial matrix expansion and arteriolar hyalinosis. Moreover, other non-inflammatory factors related to kidney dysfunction include metabolic acidosis, renin-angiotensin-aldosterone system, uremic toxins, and vitamin D deficiency.

With a rich knowledge of the pathophysiological of metabolic syndrome and diabetes, there were still suboptimal effective prevention, treatment, and intervention practices for patients with established sarcopenic obesity and CKD. Although myostatin inhibitors and AST-120 have been shown to increase muscle mass in mice studies [49,50], there were concerns of inadequate evidence of these inhibitors used in muscle wasting disorders in humans or muscle hypertrophy in athletic performance. Low serum 25(OH)D levels appear to be associated with obesity, insulin resistance, low skeletal muscle mass and rapid decline of kidney function. Systemic L-carnitine depletion has been another causative factor for sarcopenia. The benefits and thresholds of nutritional supplements for vitamin D and L-carnitine deficiency are highlighted with ongoing clinical trials and consensus. Fortunately, both sarcopenic obesity and physical performance, e.g., grip strength and walking speed, are positively modified by lifestyle interventions, such as nutritional modification and exercise training in renal patients [51]. A randomized controlled trial suggests that a supervised and home-based training phase is effective, adhered to, and safe in patients with kidney disease [52]. There should be many more revised standards of the effective and feasible training intensity to increase adherence in patients with CKD.

## 7. Conclusions

Based on the adequate definition of sarcopenic obesity, awareness of these concepts and complex metabolic pathways, lifestyle modification, physical activity and weight reduction per se lead to physical fitness and preservation of muscle mass. The most effective strategies to deal with the future debilitating complications of diabetic nephropathy are preventing hyperglycemia, the early diagnosis of kidney disease, starting treatment with antihypertensive drugs that reduce the activity of the renin-angiotensin system and lipid-modifying therapy. The adequate analysis of body composition is also crucial to assess the effects of novel medications, nutrition and physical exercise interventions for metabolic disturbances. However, further confirmatory evidence and research combining different approaches are required.

## Figures and Tables

**Figure 1 life-12-00380-f001:**
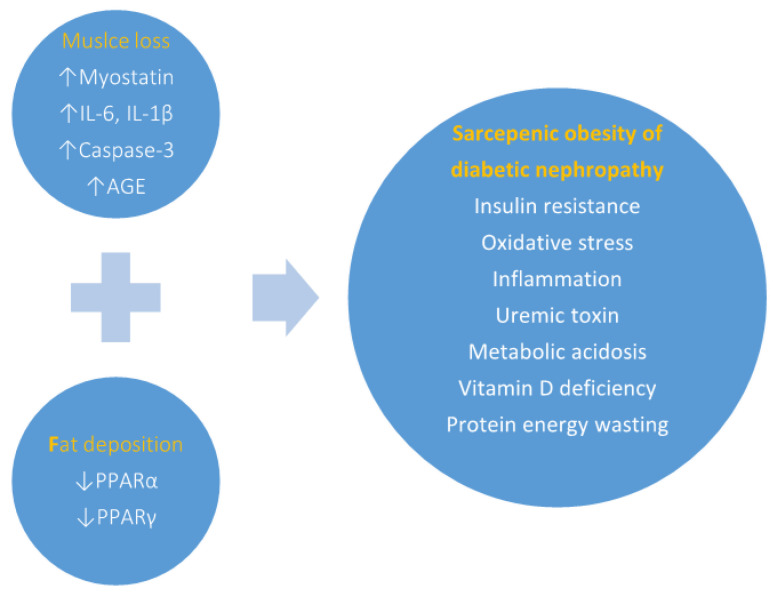
The pathophysiology of “Sarcopenic obesity of diabetic nephropathy”. Myostatin and caspase-3 overexpression in advanced CKD might contribute to protein degradation in the skeletal muscle by ubiquitin proteasome system (UPS). Both proinflammatory cytokines IL-1β and IL-18 are then secreted following the increased activity of caspase-1 and upregulation of NLRP3 inflammasome. Hyperglycemia is correlated with advanced glycosylated end-products (AGE) and IL-6, which lead to muscle wasting. Lipid accumulation interacts with decreased microenvironment signals of PPARα and PPARγ and accelerates diabetic nephropathy. Several non-inflammatory factors promote protein energy wasting in kidney disease, including uremic toxin, metabolic acidosis, vitamin D deficiency.

**Figure 2 life-12-00380-f002:**
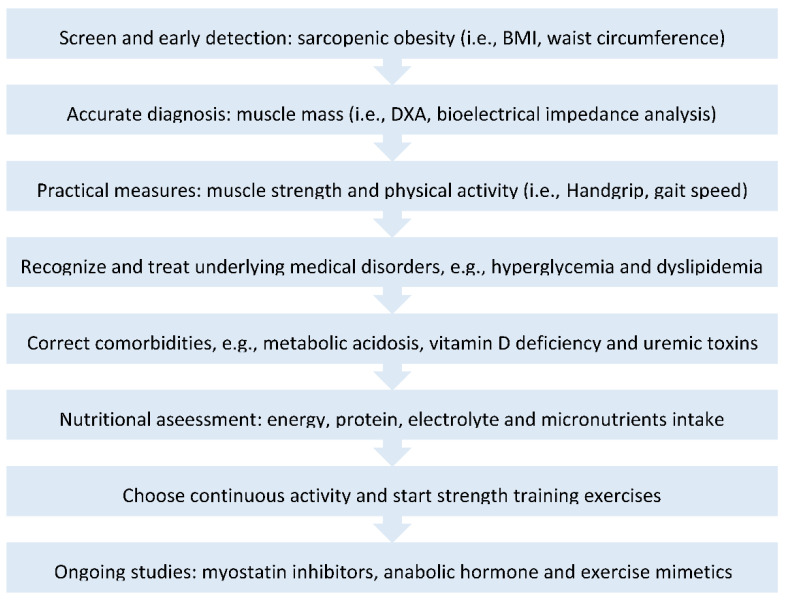
The management of “Sarcopenic obesity of diabetic nephropathy”. Accordingly, the early detection and proper diagnosis of diabetic kidney disease (DKD) and sarcopenic obesity should be implemented by routine health examinations and dual-energy X-ray absorptiometry (DXA) scans. The clinical assessment of physical performance of the upper and lower extremities function through brief tests of gait speed, time to rise from a chair five to ten times and handgrip. Low muscle mass and strength are related with comorbidities (e.g., metabolic acidosis and vitamin D deficiency) in CKD. Healthy nutrition and physical activity counteract with sarcopenic obesity. New pharmacologic therapy targeting muscle wasting and obesity are ongoing, including myostatin inhibitors, anabolic hormone and exercise mimetics.

## Data Availability

Not applicable.

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
