# Peer review of "The Role of Lifestyle Intervention, in Addition to Drugs, for Diabetic Kidney Disease with Sarcopenic Obesity"

_life, 2022, doi:10.3390/life12030380_

Round 1

Reviewer 1 Report

The paper " Beside drugs, lifestyle intervention for diabetic kidney disease patients with sarcopenic obesity" is a well written review about the relationship between lifestyle modifications and siabetic kidney damage.

Commets

Abstraction is well. Keywords are relevant.

Background information is adequate.

Diabetic kidney damage is associated with inflammation and should be emphasized. Various inflammatory markers have been suggested to be associated with diabetic kidney disease ((https://pubmed.ncbi.nlm.nih.gov/32892687/), (https://pubmed.ncbi.nlm.nih.gov/35081502/), (https://pubmed.ncbi.nlm.nih.gov/29263458/), (https://pubmed.ncbi.nlm.nih.gov/30369376/), (https://pubmed.ncbi.nlm.nih.gov/34626587/), (https://pubmed.ncbi.nlm.nih.gov/33553019/), (https://pubmed.ncbi.nlm.nih.gov/31615320/), and (https://pubmed.ncbi.nlm.nih.gov/34497035/)). I also suggest emphasizing the role of inflammation in frailty. Recent reports revealed close association between them (https://pubmed.ncbi.nlm.nih.gov/33502879/) and (https://pubmed.ncbi.nlm.nih.gov/35070222/)). Inflammation, as a common pathway, may link frailty and diabetic kidney disease.

Authors mentioned that "Besides, a cross-sectional cohort study demonstrated 25-hydroxyvitamin D (25(OH)D) serum concentration is inversely associated with several inflammatory bi-omarkers, such as high-sensitive C-reactive protein (hsCRP), TNF-α and IL-6, in severe obese individuals with mean BMI of 43.6 ± 4.3 kg/m2 ". I am totally agree with the authors since reduced vitamin D levels were associated with inflammation (https://pubmed.ncbi.nlm.nih.gov/35057450/), and type 2 diabetes mellitus (https://pubmed.ncbi.nlm.nih.gov/35157688/). In addition, low vitamin D levels contribute to sarcopenia in frail subjects (https://pubmed.ncbi.nlm.nih.gov/33029248/).

References are listed in accordance with the journal's instructions. However, seventeen of them are older than 10 years. If possible, please replace them with novel ones.

Overall, I recommend revision of the issues as mentioned above before reconsideration for publication.

Author Response

Dear Reviewer,

Thank you for taking the time to review our work. In the review, we have modified the manusrcipt according to the comments and suggestions below. We have also updated the literature search. 

Comments from the reviewers

Reviewer: 1

Comments: 

1. Abstraction is well. Keywords are relevant. Background information is adequate.

Response: Thank you very much !

2. Diabetic kidney damage is associated with inflammation and should be emphasized. Various inflammatory markers have been suggested to be associated with diabetic kidney disease. I also suggest emphasizing the role of inflammation in frailty. Recent reports revealed close association between them. Inflammation, as a common pathway, may link frailty and diabetic kidney disease.

Response: We have added new research about various inflammatory markers in first paragraph of 2. We discuss about the interactions among the NLRP3 inflammasome and diabetic kidney disease. We also updated knowledge on NLR and PLR as novel inflammation markers of frailty and its complications in second paragraph of 2.

3. Authors mentioned that "Besides, a cross-sectional cohort study demonstrated 25-hydroxyvitamin D (25(OH)D) serum concentration is inversely associated with several inflammatory bi-omarkers, such as high-sensitive C-reactive protein (hsCRP), TNF-α and IL-6, in severe obese individuals with mean BMI of 43.6 ± 4.3 kg/m2 ". I am totally agree with the authors since reduced vitamin D levels were associated with inflammation and type 2 diabetes mellitus. In addition, low vitamin D levels contribute to sarcopenia in frail subjects.

Response: We have added the sentence so it read: "Low vitamin D level is an independent predictor for sarcopenia with low handgrip strength tests in elderly men. Early screening and replacement of vitamin D in frailty need to be further investigated [28]."

4. References are listed in accordance with the journal's instructions. However, seventeen of them are older than 10 years. If possible, please replace them with novel ones.

Response: We revised older citation in the first paragraph of 5: "Another randomized placebo-controlled trial of 50 mg/day of dehydroepiandrosterone supplementation for 1 year to men and women aged 60–80 years, it failed to induce beneficial effects either on muscle strength or in muscle and fat cross-sectional areas in healthy subjects." and added a new reference as "Results similar to another randomized placebo-controlled trial of 5 mg/d of transdermal testosterone gel or placebo in older, frail men for 12 to 24 months, there was an increase in lean mass and a decrease in fat mass in the testosterone group but no available effects in strength or physical performance." We also modified the sentence of "Low dietary protein, which results in a proportional reduction in urea generation and intraglomerular pressure, is routinely proposed." 

Reviewer 2 Report

This manuscript is a comprehensive review, which focuses on the diagnosis and therapy of diabetic kidney disease associated with sarcopenic obesity. The topic of the review is clinically important and the relevant findings of the literature are discussed in a logical order; however, some critical elements are missing or require revision.

Specific suggestions:

  1. The title lacks a predicate.
  2. Figure 2 is missing.
  3. The legend(s) of the Figure(s) should be more detailed. All abbreviations within a Figure should be defined in the corresponding legend.
  4. Some parts of the manuscript text (e.g. second paragraph of 1., third paragraph of 4.) lacks references.
  5. Skeletal muscle secretes myokines but not adipokines; please, revise on page 2.
  6. Independent studies reported that PPARγ and IL-6 are strong inducers of adipocyte browning (even in humans), which has several metabolic benefits. IL-6 is not only a pro-inflammatory cytokine but also a myo- and batokine. This should be discussed in the manuscript.
  7. All abbreviations should be given when first mentioned in the text.
  8. The Authors should carefully check the grammar and spelling of words throughout the entire manuscript text.

Author Response

Dear Reviewer,

Thank you for taking the time to review our work. In the review, we have modified the manusrcipt according to the comments and suggestions below. We have also updated the literature search. 

Comments from the reviewers

Reviewer: 2

Suggestions: 

  1. "The title lacks a predicate."  Response: We modified the title to "Apart from Drugs, The Role of Llifestyle intervention Intervention for diabetic Diabetic kidney Kidney disease Disease patients with Sarcopenic Obesity"
  2. "Figure 2 is missing."  Response: We repair the figure 2 with light color sheet.
  3. "The legend(s) of the Figure(s) should be more detailed. All abbreviations within a Figure should be defined in the corresponding legend."  Response: We have added the detail legends of the figures with explaination of all abbreviations.
  4. "Some parts of the manuscript text (e.g. second paragraph of 1., third paragraph of 4.) lacks references."  Response: References were replenished according to the suggests.
  5. "Skeletal muscle secretes myokines but not adipokines; please, revise on page 2."  Response: We deleted "adipokine" of the sentence, so it reads: "It has become apparent that human skeletal muscle is an endocrine organ, which can secrete many myokines for regulation of either autocrine, paracrine or endocrine actions."
  6. "Independent studies reported that PPARγ and IL-6 are strong inducers of adipocyte browning (even in humans), which has several metabolic benefits. IL-6 is not only a pro-inflammatory cytokine but also a myo- and batokine. This should be discussed in the manuscript."  Response: We have added the discussion between Brown adipocyte and PPARγ & IL-6 in third paragraph of 4. We also presented a disussion of physiology of IL-6.
  7. "All abbreviations should be given when first mentioned in the text."  Response: We agree with the reviewer and have made a check on all abbrevations.
  8. "The Authors should carefully check the grammar and spelling of words throughout the entire manuscript text."  Response: We have checked the grammar as non-native English speakers.

Round 2

Reviewer 2 Report

The manuscript was substantially improved. Therefore, I suggest the current version of this manuscript for publication in Life.